# Modeling and Forecasting the COVID-19 Temporal Spread in Greece: An Exploratory Approach Based on Complex Network Defined Splines

**DOI:** 10.3390/ijerph17134693

**Published:** 2020-06-30

**Authors:** Konstantinos Demertzis, Dimitrios Tsiotas, Lykourgos Magafas

**Affiliations:** 1Laboratory of Complex Systems, Department of Physics, Faculty of Sciences, International Hellenic University, Kavala Campus, 65404 St. Loukas, Greece; tsiotas@aua.gr (D.T.); magafas@teikav.edu.gr (L.M.); 2Department of Regional and Economic Development, Agricultural University of Athens, Greece, Nea Poli, 33100 Amfissa, Greece; 3Department of Planning and Regional Development, University of Thessaly, Pedion Areos, 38334 Volos, Greece

**Keywords:** COVID-19 coronavirus pandemic, outbreak, modeling, prediction, regression splines, modularity optimization algorithm

## Abstract

Within the complex framework of anti-COVID-19 health management, where the criteria of diagnostic testing, the availability of public-health resources and services, and the applied anti-COVID-19 policies vary between countries, the reliability and accuracy in the modeling of temporal spread can prove to be effective in the worldwide fight against the disease. This paper applies an exploratory time-series analysis to the evolution of the disease in Greece, which currently suggests a success story of COVID-19 management. The proposed method builds on a recent conceptualization of detecting connective communities in a time-series and develops a novel spline regression model where the knot vector is determined by the community detection in the complex network. Overall, the study contributes to the COVID-19 research by proposing a free of disconnected past-data and reliable framework of forecasting, which can facilitate decision-making and management of the available health resources.

## 1. Introduction

The coronavirus disease 2019, abbreviated as COVID-19, is a contagious disease caused by the severe acute respiratory syndrome coronavirus 2 (SARS-CoV-2), which frequently causes fever, cough, and dyspnea, and, less frequently, muscle pains and neck-related problems [1,2,3]. The virus is mainly transmitted to humans through respiratory channels, and the majority of patients are asymptomatic or have soft symptoms to the disease, but in certain cases develop either pneumonia (the worst aspect of which is the fatal acute respiratory distress syndrome (ARDS)) or multi-organ deficiency [3]. The time from exposure to the appearance of symptoms ranges from 2 to 14 days, with a 5-day average, the long-range of which is affected by the relevance of the disease, in its asymptomatic or soft symptomatology aspect, with the common cold [4,5]. Around 25–30% of patients worsen after the 14th day of exposure, showing respiratory infection, whereas 83% of patients develop lymphopenia [6]. The disease is also observed in children, usually with soft symptoms [7,8].

The COVID-19 is detected either by laboratory methods, usually by the method of polymerase chain reaction (PCR) [9], where the sample is received from the rhino-laryngology region, or just by the clinical methods of evaluating the combinations of symptoms (at least of two major symptoms), danger-factors, and indications of the chest radiogram, in conjunction with the history of the patients’ contacts and movements [9,10]. Currently, since no vaccine or cure for the disease is available [11], the major efforts of the medical community are focused on the management of symptoms, while the efforts of the government are focused on the management of public health resources and the prevention management of the disease. Within the context that COVID-19 is a particularly airborne contagious disease, medical directives to the public highlight the need for careful personal body hygiene, while government policies [12] impose severe restrictions of mobility, gathering, transportation, and trade activities.

In particular, starting from mid-December 2019, when COVID-19 emerged in the city of Wuhan, China, up to 19 April 2020, the disease was spread to 210 countries, causing 2,408,123 infections and 165,105 deaths [13]. Despite that its spatiotemporal pattern differs amongst countries worldwide, it is a common feature that the pandemic shows scaling trends worldwide (with a couple of exemptions in the cases of South Korea and China) without showing a tendency of stabilization. For instance, up to 19 April 2020, North America recorded 43,369 deaths and 820,749 infections, South America 3850 deaths and 82,310 infections, Asia 14,801 deaths and 383,542 infections, Africa 1128 deaths and 22,992 infections, and Oceania 83 deaths and 8150 infections [14,15]. On the other hand, Europe was more badly affected by the pandemic. Although it accounted for half of the global infections (1,089,659), it recorded over 60% of worldwide deaths (101,859), from which 78.4% (72,196 deaths) were in Italy (23,660), Spain (20,453), France (19,718), and the United Kingdom (16,060) [14,15].

In contrast to the worldwide and European status of COVID-19, Greece has a proportion of 240 confirmed infections per million of population, which is almost 35% lower than the global average, which is about 370 infections per million, and 85% lower than the European average, which is about 1330 infections per million [14,15]. Within the context of promising a success story in the fight against the disease, this paper develops a novel nonparametric method for the modeling of the evolution of the Greek COVID-19 infection-curve, which can facilitate more accurate forecasting. The proposed method builds on a recent conceptualization of detecting communities of connectivity in a time-series [12] and develops a novel model based on the regression splines algorithm that is more accurate and reliable in forecasting. The overall approach provides insights into good policy and decision-making practices and management that can facilitate the decision-making and management of the available health resources in the fight against COVID-19.

The remainder of the paper is organized as follows: Section 2 reviews the literature in the current analysis of COVID-19 temporal spread, Section 3 applies a descriptive analysis of COVID-19 in Greece, Section 4 describes the methodological framework of the proposed method, Section 5 shows the results of the analysis and discusses them within the context of public-health management and practice, and, finally, in Section 6, conclusions are given.

## 2. Literature Review

The work of [16] is a detailed presentation of COVID-19 records that were extracted from national, regional, and municipal health-reports, and web information, aiming to contribute to decision-making for public health with insightful primary information. This work focuses more on the recording than on the analysis of cases, and, therefore, it exclusively contributes to COVID-19 research as an archive of statistical data. The work of [17] is an insightful time-series analysis examining the interconnection between deaths and infected cases, based on four health indicators of COVID-19, in China. The analysis uses cross-sectional dependence, endogeneity, and unobserved heterogeneity estimation methods, and detects a linear relationship between COVID-19 attributable deaths and confirmed cases, whereas a nonlinear relationship rules the nexus between recovery and confirmed cases. This work contributes to the literature with an interesting case-study that is, by default, restricted to the case of China, to the limited number of indicators used in the analysis, and to the limited time-series dataset that does not facilitate reliable forecasting. Next, the work of [18] proposed a heuristic method for estimating basic epidemiologic parameters for modeling and forecasting the COVID-19 spread based on available epidemiologic data. Their approach suggests a reverse forecasting process that builds on spreading scenarios, which reproduce the confirmed cases, and it develops a directed tendency which cannot promise a reliable basis for forecasting. In addition, [19] studied the temporal-spread of the disease based on exponential smoothing modeling. Although interesting, this approach is restricted to the insufficient amount of past data on which their exponential model was based, and it promises to forecast the tendency of the COVID-19 future-spread model according to illnesses of the past, while the fitted curve is calibrated and smoothened in accordance with the foregoing available cases of other countries. Moreover, [20] presented an interesting forecasting model based on a polynomial neural network with corrective feedback, which is capable of forecasting with satisfactorily accuracy, even in cases of insufficient data availability. Although interesting, this approach should be further tested and compared with alternative established algorithms of similar good accuracy by taking into consideration more than the accuracy criterion for the comparison.

On the other hand, due to the diversity that the phenomenon has in different countries, many researchers were focused on national case-studies of COVID-19 instead of the global case. For instance, [21] demonstrated the changes in statistical data of the United Kingdom after the application of the anti-COVID-19 social distance policies. In Italy, many studies were conducted for the modeling of the pandemic [22,23,24] due to the fatal outbreak that the disease had in the country, which attracted global attention. The work of [22] is a characteristic early study of modeling the spatiotemporal spread of COVID-19 in Italy, where the analysis builds on statistical modeling but without testing the statistical significance of the research hypothesis. Within the context of epidemiologic research, this incompleteness restricts the contribution of this interesting approach, provided that in epidemiologic studies the goal is to develop an occurrence function (as a measure of association), quantifying a cause–effect relation between a determinant (cause) and its result (effect), and therefore the major concern is to test whether this cause–effect relation is statistically significant.

Greece is an example of a timely response in the application of anti-COVID-19 policies that are currently have been proven effective in keeping the infected cases and deaths at relatively low levels [12,13,16]. In particular, the first infection emerged in the country on 26 February 2020, and just three days later, the state began applying several policies for the control of the disease [12]. This timely response has led Greece to be currently considered as a successful case in anti-COVID-19 management compared to both the European and global cases [13,16]. At the time that Greece started to attract global attention, the authors of [12] proposed a novel complex network analysis of time-series, based on the visibility algorithm [25,26], for the study of the Greek COVID-19 infection curve. The authors showed that the evolution of the disease in Greece went through five stages of declining dynamics, where saturation trends (represented by a logarithmic pattern) emerged after the 33rd day (29 April 2020). Within the context that Greece promises a success story and an insightful case study, both in epidemiologic and in anti-COVID-19 policy terms, this paper builds on the very recent work of [12] and advances the time-series modeling and forecasting by developing a model based on the regression splines algorithm that is more capable of providing accurate predictions of future trends.

## 3. Descriptive Analysis of COVID-19 in Greece

The disease of COVID-19 emerged in Greece on 26 February 2020, almost two months after its global emergence [12]. As it can be observed in Figure 1, within the 54 first days of the pandemic (until 19 April), Greece recorded 2235 confirmed infected cases [15], from which 56% were men, 25.5% (570 cases) were related with traveling abroad, and 42.2% (943 cases) were linked with other confirmed cases, whereas the others were untracked and were still undergoing investigation [27]. The average age of cases was 49 years (ranging from 1 day until 102 years old), whereas the median death age was 74 years (ranging from 39 to 95 years) [27].

In numeric terms, Greece is at the 58th place in the number of infected cases worldwide, and at the 46th place in the number of deaths [15], while, in Europe, Greece is at the 25th place in the number of cases and at the 22nd place in terms of deaths. In per capita terms, Greece currently has 13 deaths per million of population [15], while the global average is above 16 deaths per million of residents [14]. Additionally, with 67 patients being under a serious-critical situation, Greece is at the 37th place worldwide, and at the 20th place in Europe [14,15,27]. In terms of testing, Greece has conducted 50,771 tests and takes the 56th place worldwide and the 23rd place in Europe [14,15,27].

The geographic distribution of the confirmed infected cases in Greece are shown in the map of Figure 2 [27], where it can be observed that the majority of infections are concentrated in the metropolitan prefectures of Attiki (6) and Thessaloniki (47), along a vertical axis configured by the prefectures of Kastorias (24), Kozanis (30), and Larissas (33) in central Greece, in the prefecture of Euvoias (12), in the prefectures of Xanthis (50) and Evrou (13) at the north-east of the country, and at the prefectures of Achaias (1), Heleias (19), and Zakenthou (51) in south-west Greece, the majority of which are transportation (road, maritime, and air transport) centers.

Additionally, Greece is in the 68th place worldwide and in the 26th place in Europe regarding the number of patients that have recovered from COVID-19 [14,15,27]. The daily and cumulative infections of COVID-19 in Greece are shown in Figure 3.

Next, Figure 4 shows the evolution of the confirmed infected cases in Greece compared to the respective recorded deaths. The vertical axis of the diagram is graded at the logarithmic scale, where linear segments illustrate the exponential growth of the disease (the slope of the linear growth is proportional to the size of the exponent). As it can be observed, the almost constant offset between the two curves implies that the number of infections and the number of deaths of COVID-19 in Greece are correlated. This interprets that these two indicators follow a similar growth pattern, which is in line with the shape of the curves shown in Figure 1 and Figure 3. In addition, a particularly promising observation is the declining growth rates shown in Figure 4 for both the infection and death curves. However, these observations will be statistically tested in the following part of the analysis.

Next, Figure 5 is an aggregate diagram showing the evolution of COVID-19 confirmed (total) cases, new infections, deaths, and recovered on official data extracted from the Greek Ministry of Health [27].

Within the context of the global outbreak of the disease, the previous descriptive analysis illustrates that Greece suggests a good example for its COVID-19-related sizes, which keep the country at the last places both in the European and the global ranking. However, this good performance has been the result of the timely response and application of anti-COVID-19 policies in Greece [12], including social distancing to prevent the spreading of the disease, In particular, the first anti-COVID-19 policies in Greece were applied after the confirmation of the first three infected cases, which were dated 27 February 2020. On that day, all carnival events were canceled to prevent an outbreak of the disease. On 10 March, the number of total cases reached 89 [28], the tracking of which revealed that they were mainly related to travelers originating from Italy or with pilgrims returned from a religious excursion to Israel with travelers from Egypt, along with their contacts [28]. On that day, the government announced the directives related to personal hygiene, social distancing, and prevention; the anti-COVID-19 measures up to then were optional and applicable at the local level (and, particularly, at the regions with infected cases such as Heleias-19, Achaias-1, and Zakynthou-51) and mainly concerned the local suspension of schools, school excursions abroad, and cultural events [12]. However, on 10 March, due to the spreading of the disease to multiple regions and due to the disobedience of the citizens to conform with the measures, the government applied more active measures and proceeded to the national suspension of all educational structures (at all ranks), and a couple of days later, on 12 and 13 March, it proceeded to suspend cafeterias, bars, museums, malls and trade centers, sports activities, and restaurants [28]. On 16 March, all commercial shops were suspended at the national level, two villages in the regions of Kozani (30) were put into quarantine, and all doctrine and religious activities were suspended [28]. The only active businesses and firms exempted from these measures were primary need suppliers, such as bakeries, supermarkets, pharmacies, and private health services [28]. Aiming to support the anti-COVID-19 policy of social distancing, the government announced, on 18 and 19 March, a 10-billion Euros (€) budget for taxation benefits, regulations, or subsidies for the support of the economy, companies, and workers affected by the suspensions and social distancing [28]. On 23 March 2020, the government announced national restrictions in transportation, with exemptions for commuting to work, movements for supplies of food, medicines, medical services, and health (gyms). However, these exemptions had to be documented by identification papers, such as ID-cards or passports, with an affirmation paper stating the purpose of movement [28]. Provided that the citizens are within the accepted exceptions, the citizens must carry both police identity or passports, as well as some type of certification as to the purpose of travel. This measure was applicable until 27 April 2020, and it was extended until 4 May 2020 [28].

In the fight against the disease, the development of more accurate and reliable models in terms of description and prediction can help policymakers better conceptualize the pandemic and apply proper and more effective policies. Towards this direction, this paper proposes a novel complex-network-based approach of the splines algorithm, which facilitates better epidemiologic modeling and forecasting.

## 4. Methodology and Data

The available data were extracted from the National Public Health Organization of Greece [27] and the Ministry of Health of Greece [28]. The variables participating in the analysis include daily cases of the period 26 February 2020 until 16 April 2020 and are the day since the first infection in Greece (variable X_1_: Day), the COVID-19 cumulative infected cases (variable X_2_: Infections) and cumulative deaths (var. X_3_: Deaths), the daily infections (var. X_4_: Daily Infections), the daily deaths (var. X_5_: Daily Deaths), daily recoveries (var. X_6_: Daily Recovered), the daily new patients in intensive care units (var. X_7_: ICU), and the daily number of tests (var. X_8_: Tests). All available variables are shown in Table A1 (Appendix A). Each variable is a time-series *x*(n) = {*x*(*i*)|*i* = 1, 2, …, *n* } = {*x*(1), *x*(2), …, *x*(*i*)}, where each node *i* = 1, 2, …, *n* refers to a day since the first infection.

Overall, the analysis examines the dynamics of the Greek COVID-19 infection curve as it is expressed by the available time-series variable X_2_:X_8_. The study is implemented through a double perspective; the first examines the structural dynamics of one variable (X*_i_*, *i* = 1, …, *n*) in comparison with the other available variables X*_j_* (analysis between variables, *i* ≠ *j* = 1, …, *n*), whereas the second examines the time-series pattern configured for a variable X*_i_* (analysis within variable X*_i_*, *i* = 1, …, *n*). Towards the first direction, Pearson’s bivariate correlation analysis is applied to the set of the available variables (X_2_:X_8_), and the results are shown in Table 1. In terms of time-series analysis [29], computing Pearson’s bivariate correlation coefficients for variables x, y is equivalent to a cross-correlation analysis with a zero lag (h = 0) applied between variables x_t_ and y_t+h_. Therefore, the first structural perspective does not build on a lagged consideration of the available data since there is neither theoretical evidence in epidemiologic studies [30,31] nor any indication based on data observation that the available time-series have periodical structure. As it can be observed in Table 1, the number of infections (X_2_) is significantly correlated with all variables except X_6_ (daily recovered) and the daily number of infections (X_4_) is significantly correlated with the daily number of deaths (X_5_) and the patients in ICU (X_7_). These significant results imply, on the one hand, that the coevolution of the COVID-19 infection curve with variables X_3_:X_5_, X_7_, X_8_ is less than 5% likely to be a matter of chance, and, on the other hand, that the coevolution of the daily COVID-19 infections with the daily number of deaths and the patients in ICU is less than 1% likely to be a matter of chance. In general, the correlation analysis indicates that the evolution of the COVID-19 infections in Greece is very likely to submit to causality and less likely to be a matter of chance.

As far as correlations of other variables are concerned, Table 1 shows that the number of recoveries (X_6_) is significantly (but not highly) and positively correlated with the number of daily deaths (X_5_), expressing a tendency of the Greek health system to get more recoveries when the number of deaths increases. This correlation illustrates the analogy between deaths and recoveries, suggesting a variable for further research. In addition, the number of patients in ICU (X_7_) appears significantly and negatively correlated with the number of infections (X_2_) and deaths (X_3_), implying that the number of patients in ICU tends to decrease when the number of infections and deaths increases. This observation is rationale since cases of death are removed from the ICU. On the other hand, the number of patients in ICU (X_7_) is significantly and positively correlated with the number of daily infections (X_4_), implying that the number of patients entering in ICU tends to increase when the number of daily infections gets bigger. Next, an interesting observation regards the correlations between the number of tests (X_4_) and variables X_2_ and X_3_. Although these correlations *r*(X_4_,X_2_) and *r*(X_4_,X_3_) are significant and positive, as expected (implying that the number of tests appears proportional to the number of infections and deaths), the numerical values of these coefficients do not appear considerably high (since *r*(X_4_,X_2_), *r*(X_4_,X_2_) < 0.6). Provided that perfect positive linearity between the number of tests and infections (or deaths) implies increasing awareness of the health system proportionally to the spread of the disease, the considerable high distance (>40%) of the correlation coefficients *r*(X_4_,X_2_) and *r*(X_4_,X_2_) from perfect (positive) linearity can be seen as an aspect of testing ineffectiveness of the health system in Greece. Overall, the correlation analysis shows that different aspects of the disease in Greece are ruled by nonrandomness, and, therefore, it provides indications that the evolution of the Greek COVID-19 system is driven by short-term linear trends. Therefore, a stochastic analysis is further applied for improvement of the overall system’s determination and, thus, for the better conceptualization of the dynamics ruling the evolution of COVID-19 in Greece.

### 4.1. Regression Analysis

The first approach for modeling the evolution of the GOVID-19 infection curve builds on regression analysis and generally on the curve fitting approach [32], according to which a parametric curve is fitted to the data of variable X_2_ = *f*(*t*) that best describes its variability through time. The available types of fitting curves examined in the regression analysis are linear, quadratic (2nd order polynomial), cubic (3rd order polynomial), power, and logarithmic. All the available types of fitting curves can be generally described by the general multivariate linear regression model expressed by the formula [32]
(1)$$\hat{y}=b_{1}x_{1}+b_{2}x_{2}+\ldots +b_{n}x_{n}+c=\sum b_{i}x_{i}+c$$
by considering that each independent variable *x_i_* can represent a function of *x*, namely, *x_i_* = *f*(*x*), as it is shown in relation (2).
(2)$$\hat{y}=\sum b_{i}f\left(x_{i}\right)+c$$

The function *f*(*x*) can be either logarithmic *f*(*x*) = (log(*x*))*^m^*, or polynomial *f*(*x*) = *x^m^*, or exponential *f*(*x*) = (exp{*x*})*^m^*, or any other. Within this context, the purpose of the regression analysis is to estimate the parameters *b_i_* of Model (2) that best fits the observed data y, so that to minimize the square differences of $y_{i}-\hat{y_{i}}$ [32], namely:(3)$$min\left\{e=\overset{n}{\underset{i=1}{\sum }}{\left[y_{i}-\hat{y_{i}}\right]}^{2}=\overset{n}{\underset{i=1}{\sum }}{\left[y_{i}-\left(\sum b_{i}f\left(x_{i}\right)+c\right)\right]}^{2}\right\}$$

The algorithm estimates the beta coefficients (*b_i_*) by using the least-squares linear regression (LSLR) method [32] based on the assumption that the differences *e* in Relation (3) follow the normal distribution *N*(0,$\sigma_{e}^{2}$). In this paper, the time (days since the first infection, variable X_1_) is set as an independent variable, and each other available variable is set as a response variable to the models. In all cases, the simplest form of regression model that best fits the data is chosen. The simplicity criterion regards both the number of the used terms $b_{i}f\left(x_{i}\right)$ and the polynomial degree *m*. That is, the model with the least possible terms and the lowest possible degree *m* < *n−2* (where *n* is the number of observations in the dataset) is chosen if it best fits the data. The determination ability of each model is expressed by the coefficient of determination *R*^2^, which is given by the formula [32,33]:
(4)$$R^{2}=1-\frac{\sum_{i=1}^{n}{\left(Y_{i}-{\hat{Y}}_{i}\right)}^{2}}{\sum_{i=1}^{n}{\left(Y_{i}-{\bar{Y}}_{i}\right)}^{2}}$$
where $\;Y_{i}\;$ are the observed values of the response (dependent) variable, ${\hat{Y}}_{i}$ are the estimated values of the response variable, $\bar{Y}$ is the average of the observed values of the response variable, and *n* is the number of observations (the length of the variables). The coefficient of determination expresses the variability of the response variable, as expressed by the model $\hat{Y}$, and ranges within the interval [0, 1], showing perfect determination when it equals to one [32,33].

Another measure of fitting ability is the root mean square deviation or error (RMSD or RMSE), which calculates the square root of the expected differences between the predicted ($\hat{y}$) and the observed (*y*) values of the model, according to the formula [32,33]:(5)$$RMSE=\sqrt{MSE\left(\hat{y}\right)}=\sqrt{E\left({\left(\hat{y}-y\right)}^{2}\right)}$$
where *E*(·) is the function of the expected value. The RMSE represents the square root of the second sample moment of the regression residuals [32,33].

A final measure of fitting ability used in the analysis is the relative absolute error (RAE), which calculates the relative value of the RMSE in accordance with the expected observed values, as is shown by the formula [32,33]:(6)$$RMSE=\sqrt{E\left({\left(\hat{y}-y\right)}^{2}\right)}/\sqrt{E\left({\left(y\right)}^{2}\right)}$$
where *E*(·) is the function of the expected value. The RAE is often used in machine learning, data mining, and operations management applications, and it represents the analogy of the RMSE relative to the expected value of the observed values.

Within this context, Figure 6 shows the results of the regression analysis applied to the (cumulative) number of infections (dependent variable: X_2_). As it can be observed, the 3rd order polynomial (cubic) fitting curve best describes the data of the Greek COVID-19 cumulative infections. The last (very recent) part of the cubic curve appears convex, implying that the number of cumulative infections tends to saturate.

Next, Figure 7 shows the results of the regression analysis applied to the (cumulative) number of deaths (dependent variable: X_4_). As can be observed, similar to variable X_2_, the 3rd order polynomial (cubic) fitting curve best describes the data of the Greek COVID-19 cumulative deaths. The shape of this curve also implies that the number of cumulative infections tends to saturate. 

Finally, Figure 8 shows the results of the regression analysis applied to the cumulative number of patients in ICU (dependent variable: cumulative X_7_). Similar to variables X_2_ and X_4_, the 3rd order polynomial (cubic) fitting curve best describes the data of (cumulative) variable X_7_, and the shape of the curve implies that the number of cumulative ICU patients also tends to saturate.

The regression analysis has shown that the best fit for the cumulative expressions of the COVID-19 infection (X_2_), death (X_4_), and ICU patients (X_7_) curves in Greece is to the 3rd order polynomial (cubic) pattern than to linear, power, logarithmic, or 2nd order polynomial patterns. As was previously observed, the cubic-shape of the fitting curves (which ends up as a convex area representing the recent past of the time-series) illustrates saturation trends of the COVID-19 evolution in Greece. To improve the accuracy and determination ability of the fittings, we apply next a regression analysis based on the regression splines algorithm.

### 4.2. Regression Splines

A regression spline is a special piecewise polynomial function defined in parts, which is widely used in interpolation problems requiring smoothing. In particular, for a given partition *a* = *t*_o_ < *t*_1_ < *t*_2_ < … < *t_k_*_−1_ < *t_n_* = *b* of the interval [*a*,*b*], a spline is a multi-polynomial function *S*(*t*) defined by the union of functions [34]:(7)$$S\left(t\right)=S_{1}\left(\left[a,t_{1}\right]\right)\cup S_{2}\left(\left[t_{1,}t_{2}\right]\right)\cup \ldots \cup S_{k-1}\left(\left[t_{k-2,}t_{k-1}\right]\right)\cup S_{k}\left(\left[t_{k-1,}b\right]\right)=\cup_{i=1}^{k}S_{i}\left(\left[t_{i-1},t_{i}\right]\right),$$
where *k* is the number of knots *t* = (*t*_o_, *t*_1_, *t*_2_, …, *t_n_*) dividing the interval [*a*,*b*] into *k*−1 convex subintervals. Each function *S_i_*(*t*), *i* = 1, …, *n*, is a polynomial of low (usually square) degree (sometimes can also be linear) that fits to the corresponding interval [*t_i_*_−1,_
*t_i_*], *i* = 1, …, *n*, so that the aggregate spline function is continuous and smooth. The spline algorithm is preferable than that of simple regression in cases when the simple regression generates models of high degree [34]. This piecewise approach yields models of high determination by using low degree polynomial piece-functions. In terms of the bias-variance trade-off dilemma [35], stating that simple (i.e., of low degree) models have small variance and high bias whereas complex models have small bias and high variance, the spline algorithm can generate fittings of both low variance and low bias, and thus it minimizes the expected loss expressed by the sum of square bias, variance, and noise.

For global high degree polynomials, in order to avoid that the tail wags a lot, two extra constraints have been added at the boundaries (on each end). The constraints make the function extrapolate linearly beyond the boundary knots. With these constraints, the function goes off linearly beyond the range of the data. They then free up a few parameters, so the degrees of freedom are always the number of terms that go into the formula minus one (the intercept). The degrees of freedom are, therefore, the number of predictors that have non-zero coefficients in the model. The spline with *K* knots has *K* degrees of freedom because we get back two degrees of freedom for the two constraints on each of the boundaries [34].

The major modeling choices for applying splines are, first, the determination of the knot vector *t* = (*t*_o_, *t*_1_, *t*_2_, …, *t_n_*) so as to obtain the smoothest and best determination spline model, and, secondly, the selection of the polynomial degree, so that the model is smooth and continuous at the borders of the subintervals. Therefore, this highly effective (in terms of model determination) fitting method is very sensitive to the selection of the knot vector, which is usually being determined either uniformly, or arbitrarily, or intuitively, or based on the researchers’ experience [34,35]. The more sophisticated knot-selection techniques in the literature [36,37] build on heuristics to determine the knot vector, generating the best fitting and smoothening of the spline model. Within this open debate of knot determination, this paper builds on the recent work of [12] and introduces a novel approach for the determination of the spline knot vector, based on complex network analysis. More specifically, the proposed model introduces a novel approach for the determination of the spline knot vector based on complex network analysis based on the COVID-19 infection curve of Greece. According to this approach, the spline is divided into five knots that represent the evolution of the disease in Greece, which went through five stages of declining dynamics [12].

### 4.3. Complex Network Analysis of Time-Series

Transforming a time-series to a complex network is a modern approach that recently became popular with the emergence of network science in various fields of research [26,38,39]. The most popular method to transform a complex network to a time-series is the visibility graph algorithm that was proposed by [25], which became dominant due to its intuitive conceptualization. In particular, the rationale of creating a time-series to a complex network (visibility graph) builds on considering the time-series as a path of successive mountains of different height (each representing the value of the time-series at the certain time). In this time-series-based landscape, an observer standing on a mountain can see (either forward or backwards) as far as no other mountain obstructs its visibility. In geometric terms, a visibility line can be drawn between two points (nodes) of the time-series if no other intermediating node is higher than this pair of points and obstructs their visibility [12,25]. Therefore, two time-series nodes can enjoy a connection in the associated visibility graph if they are visible through a visibility line [25]. The visibility algorithm conceptualizes the time-series as a landscape and produces a visibility graph associated with this landscape [26]. The associated (to the time-series) visibility graph is a complex network where complex network analysis can be further applied [12,26].

Within this context, by transforming the time-series of the COVID-19 infection curve to a visibility graph, we can study the time-series as a complex network. This allows the division of the visibility graph of the COVID-19 infection into connective communities based on the modularity optimization algorithm of [40]. This algorithm is heuristic and separates a complex network into communities, which are dense within (i.e., links inside the communities are the highest possible) and sparse between (i.e., links inside the communities are the highest possible) [12,26,40,41,42]. Therefore, the most distant nodes within each community can define the knots for applying the spline algorithm. This complex-network-based definition (i.e., community detection based on modularity optimization) of the knot vector offers the missing conceptualization to the splines knots, defining them as borderline points of connectivity of the modularity-based communities. According to this approach, the visibility graph of the COVID-19 infection curve is divided into five modularity-based communities, which correspond to the periods Q1 = [1, 4] ∪ [9, 19], Q2 = [5, 8], Q3 = [20, 26], Q4 = [27, 32], and Q5 = [33, 43], as it is shown in Figure 9, where positive integers in these intervals are elements of variable X_1_.

Consequently, the spline knot vector can be defined by the knots *t* = (1, 4, 8, 19, 26, 32, 43) in the body of the time-series COVID-19 infection curve. This partition facilitates the application of the spline regression algorithm and comparison of the determination ability of the spline model with the cubic regression models previously shown.

## 5. Results and Discussion

After the complex-network-based determination of the spline knot vector, the spline regression algorithm is applied to the COVID-19 infection curve. The results are shown in Table 2, in comparison with the cubic fittings and with regression splines of randomly selected (3, 4, and 5) knots. As can be observed, in all cases (i.e., for the dependent variables X_2_, X_4_, and X_7_), the proposed complex-network spline models have better determination ability and lower error terms than both the cubic models resulted by the regression analysis and the randomly calibrated splines. In particular, improvements caused by the proposed method range between 0.00–0.20% for the multiple correlation coefficients (*R*), between 0.10–0.51% for the model determination (*R*^2^), between 0.37–41.32% for the root mean square error (RMSE), and 0.25–34.19% for the relative absolute error (RAE). These improvements are considerable even in the cases of *R* and *R*^2^, given the already good fitting performance of the cubic and randomly calibrated spline models. Additionally, it provides the area under the ROC curve metric (AUC) in order to assess the model performance [43]. A ROC curve (receiver operating characteristic curve) is a graph showing the performance of a model at all thresholds. AUC measures the entire two-dimensional area underneath the entire ROC curve from (0,0) to (1,1).

According to these results, the proposed complex-network-based splines regression method outperforms the fitting determination of both the cubic regression and the randomly calibrated splines regression models, which are also models of high accuracy. In conceptual terms, this outperformance may be related to the immanent property of complex network analysis to model and manage problems of complexity and thus to provide better insights in the study complex systems, as in the case of the COVID-19 temporal spread. Despite the restriction in data availability, improvements (mainly in error terms) achieved by the proposed model are not negligible and highlight the direction of using hybrid or combined methodologies for dealing with cases of insufficient information. Moreover, the overall approach highlights the utility of multidisciplinary and synthetic modeling for dealing with complexity in epidemiology, which, by default, deals with complex socio-economic systems of humanity.

Given the small data availability, this improvement in the modeling determination is a promising advantage of the proposed method, which allows us to build on a quantitative consideration of the spline knot vector instead of an intuitive one. Moreover, the utility and effectiveness of the proposed methodology should be evaluated in conjunction with the good performance of the spline results against multicollinearity [32,33], which emerges in simple regression modeling due to the consideration of additional parameters.

Overall, the two major advantages of the proposed method concern the better stability and capability of forecasting, since the total behavior of the proposed model appears less noisy, while it reduces the errors of determination due to the modelers’ choices. This conclusion can also be supported by the variance minimization of the expected error terms, which provides strong indications of the system’s reliability.

## 6. Limitations and Further Research

This study is based solely on the COVID-19 data from Greece, which has been relatively effective in containing the spread of the virus and where ICU admissions and deaths have been restricted to relatively low proportions of the infected population. The application of the model proposed has not yet been tested in other countries and may, therefore, need further examination or fine-tuning to adapt to the new data. Suggestions for the evolution and future improvements of this method should focus on tests on data from other countries that have been more badly affected by the pandemic, such as Italy and Spain. Additionally, it is quite interesting to see the spread of the virus between countries in terms of efficiency of the enforcement of the rules of social distancing, quarantine, and isolation. On the other hand, future research could focus on further optimization of the hyper-parameters of the algorithms used in the proposed architecture. This may lead to an even more efficient, more accurate, and faster process. Finally, an additional element that could be considered in the direction of future expansion concerns the operation of the network by means of self-improvement and redefinition of its parameters automatically. It will thus be able to fully automate the process of extracting useful intermediate representations from new time-series datasets.

## 7. Conclusions

Accurate forecasting is a major task in epidemiology that becomes very important today in the global emergence of the COVID-19 pandemic. Due to the low availability of data, the worldwide conceptualization of the new pandemic is currently constrained and still emerging. Within the context of information-lack, methods contributing to more accurate forecasting on early datasets are welcomed and pertinent for the ongoing fight against the disease. This paper proposes a novel method for modeling and forecasting in epidemiology based on complex network analysis and the spline regression algorithm. Based on data of the COVID-19 temporal spread in Greece, the proposed method converted a time-series to an associated visibility graph, and then it divided the graph into connected communities that defined the spline knot vector. This approach provided a complex-network definition of the spline knots, the definition of which is currently either intuitive or heuristic, and it assigned a conceptualization to the knots based on network connectivity.

Within this context, the proposed method was applied to different aspects of the COVID-19 temporal spread in Greece (the cumulative number of infections, deaths, and ICU patients) and was found to outperform the regression cubic models, which had the highest determination amongst the available simple regression models. In methodological terms, the overall approach advances the spline regression algorithm, which is currently restricted to the not-well-defined determination of knots, whereas, in practical implementation, the proposed methodology offers an active method for modeling and forecasting the pandemic, capable of removing disconnected past data from the time-series structure. On the effectiveness of imposing restrictive measures in a graded self-organized criticality epidemic spread model [44], and more specifically in terms of management of health policy [12], this paper provides a modeling and forecasting tool that facilitates decision making and resource management in epidemiology, which can contribute to the ongoing fight against the pandemic of COVID-19.

## Figures and Tables

**Figure 1 ijerph-17-04693-f001:**
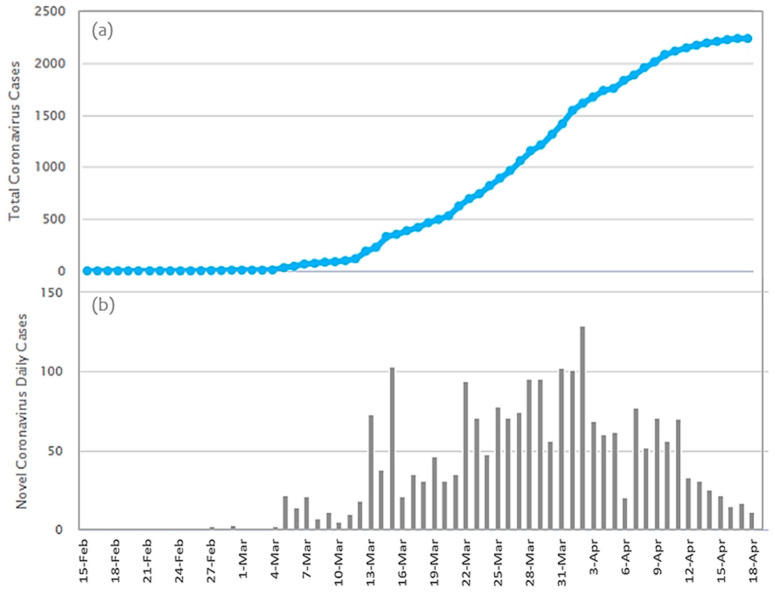
The time-series of the COVID-19 infection curve in Greece for the period 15 February 2020 to 18 April 2020. The first infection emerged on 26 February 2020 (and was recorded on 27 February 2020; data source [15]). (**a**) Total Coronavirus Cases; (**b**) Novel Coronavirus Dairy Cases.

**Figure 2 ijerph-17-04693-f002:**
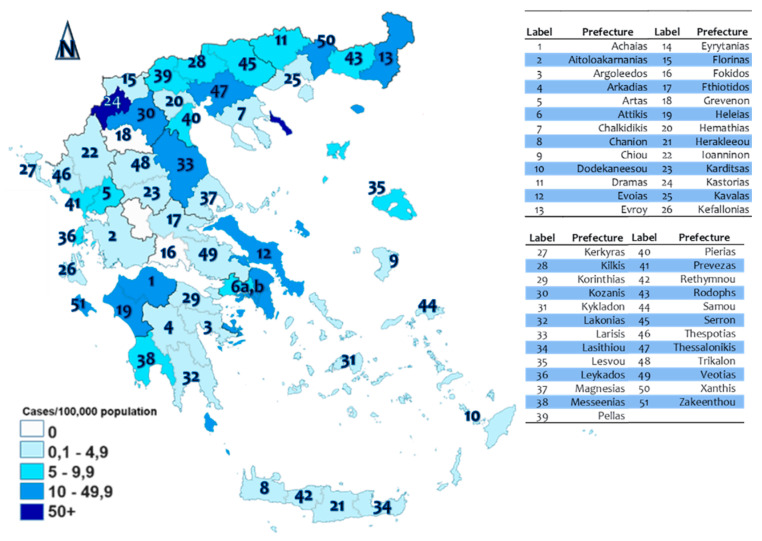
Infected cases per million in Greece (source: [27]).

**Figure 3 ijerph-17-04693-f003:**
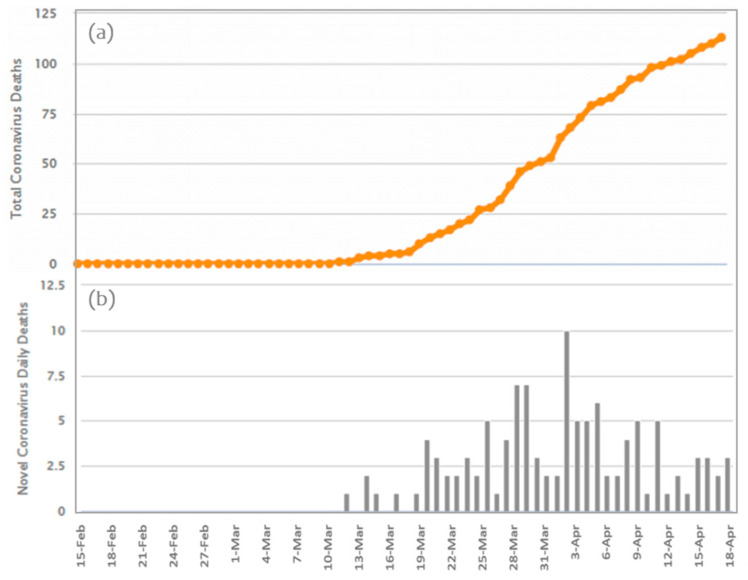
The time-series of the COVID-19 death curve in Greece for the period 15 February 2020–18 April 2020. The first death was recorded on 12 March 2020 (data source: [15]). (**a**) Total Coronavirus Deaths; (**b**) Novel Coronavirus Dairy Deaths.

**Figure 4 ijerph-17-04693-f004:**
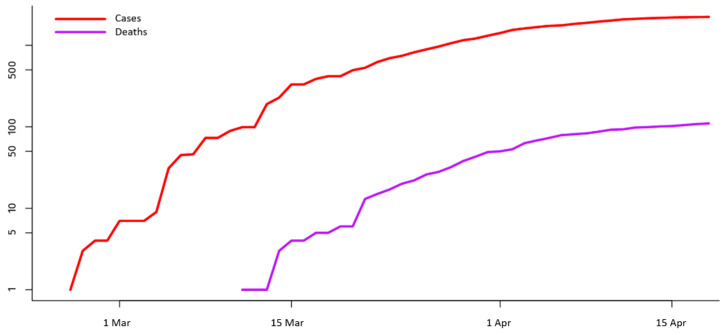
Comparative diagram with the time-series of the COVID-19 infection cases versus the recorded deaths in Greece (data source: [27]).

**Figure 5 ijerph-17-04693-f005:**
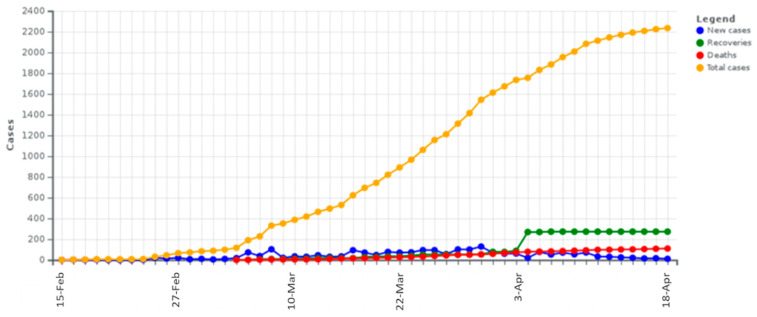
The aggregate time-series of the COVID-19 in Greece, showing the number of confirmed (total) cases, new infections, deaths, and recoveries, for the period 15 February 2020–18 April 2020 (data source: [15]).

**Figure 6 ijerph-17-04693-f006:**
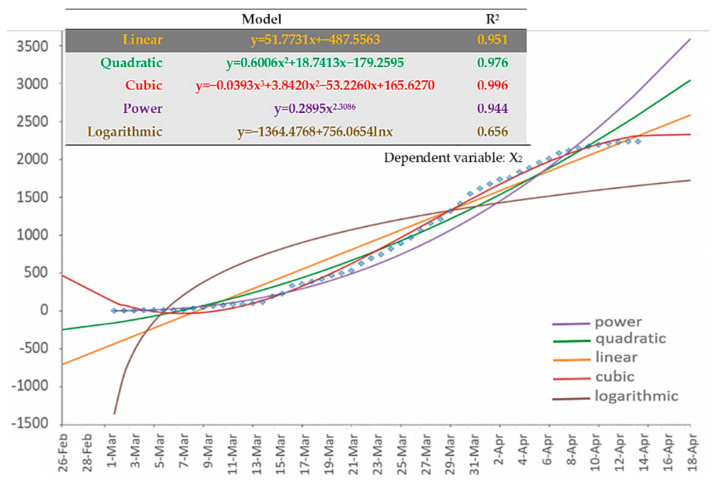
Available types of fitting curves applied to the cumulative COVID-19 infection curve (variable X_2_) of Greece. Time-series data of the variable are shown in dots.

**Figure 7 ijerph-17-04693-f007:**
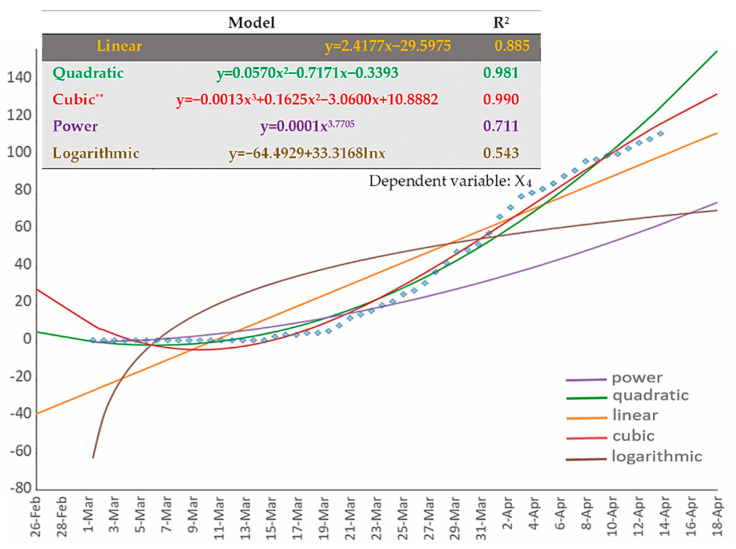
The available types of fitting curves applied to the cumulative COVID-19 death curve (variable X_4_) of Greece. Time-series data of the variable are shown in dots.

**Figure 8 ijerph-17-04693-f008:**
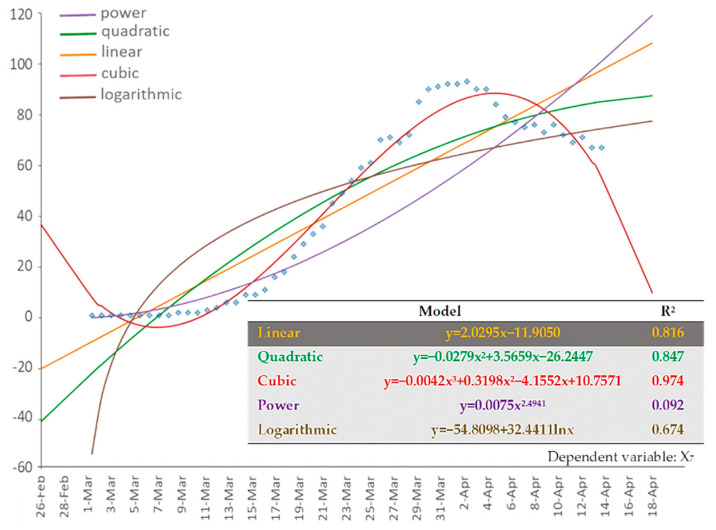
The available types of fitting curves applied to the cumulative COVID-19 ICU patients (variable: cumulative X_7_) of Greece. Time-series data of the variable are shown in dots.

**Figure 9 ijerph-17-04693-f009:**
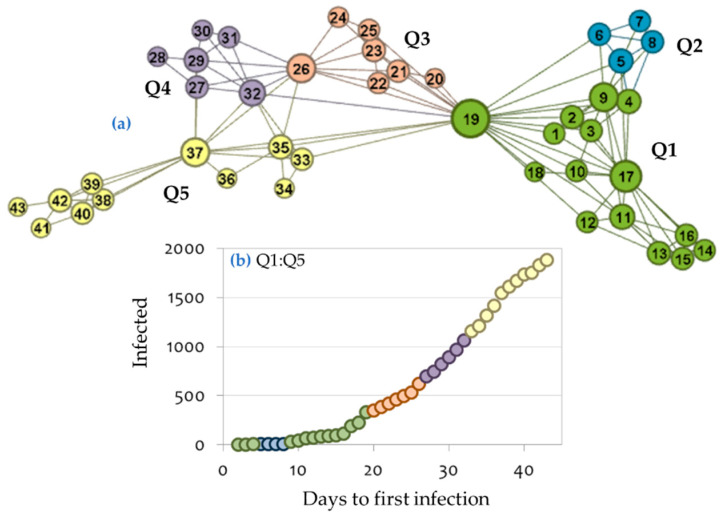
Community detection of the Greek COVID-19 infection visibility graph based on the modularity optimization algorithm of [40]. Node size in the network is proportional to node degree.

**Table 1 ijerph-17-04693-t001:** Results of Pearson’s bivariate correlation analysis (r) ^a^.

		X_3_	X_4_	X_5_	X_6_	X_7_	X_8_
Infections (X_2_)	r(x,y)	0.979 **	0.317 *	0.612 **	0.138	−0.316 *	0.585 **
	Sig. (2-tailed)	0.000	0.020	0.000	0.321	0.020	0.000
Deaths (X_3_)	r(x,y)	1	0.153	0.496 **	0.123	−0.421 **	0.552 **
	Sig. (2-tailed)		0.268	0.000	0.376	0.002	0.000
Daily Infections (X_4_)	r(x,y)		1	0.487 **	−0.038	0.358 **	0.232
	Sig. (2-tailed)			0.000	0.785	0.008	0.091
Daily Deaths (X_5_)	r(x,y)			1	0.277 *	0.010	0.339 *
	Sig. (2-tailed)				0.042	0.941	0.012
Recovered (X_6_)	r(x,y)				1	−0.133	0.003
	Sig. (2-tailed)					0.339	0.982
ICU (X_7_)	r(x,y)					1	−0.077
	Sig. (2-tailed)						0.582

^a^ This analysis equivalents to a cross-correlation analysis with a zero lag (h = 0) applied between variables x_t_ and y_t+h._ * Coefficient is significant at the 0.05 level. ** Coefficient is significant at the 0.01 level.

**Table 2 ijerph-17-04693-t002:** Comparison between the polynomial cubic and regression splines fitting curves.

Model	R	R^2^	RMSE *	RAE **	AUC
**Dependent Variable: Infections (X_2_)**					
Cubic	0.998	0.996	2.229	4.182%	0.9962
Regression Splines with 3 random knots	0.999	0.998	1.805	3.798%	0.9979
Regression Splines with 4 random knots	0.999	0.998	1.621	3.277%	0.9984
Regression Splines with 5 random knots	0.999	0.998	1.420	2.986%	0.9985
Complex-Network Regression Splines	1.000	1.000	1.308	2.752%	0.9998
**Dependent Variable: Deaths (X_4_)**					
Cubic	0.995	0.990	2.423	4.981%	0.9903
Regression Splines with 3 random knots	0.995	0.990	2.584	5.013%	0.9904
Regression Splines with 4 random knots	0.995	0.990	2.423	4.798%	0.9903
Regression Splines with 5 random knots	0.995	0.990	2.410	4.732%	0.9905
Complex-Network Regression Splines	0.995	0.991	2.401	4.720%	0.9912
**Dependent Variable: ICU Patients (X_7_)**					
Cubic	0.987	0.974	6.300	12.659%	0.9743
Regression Splines with 3 random knots	0.987	0.974	6.287	12.648%	0.9744
Regression Splines with 4 random knots	0.988	0.976	6.186	12.114%	0.9762
Regression Splines with 5 random knots	0.989	0.979	6.204	12.007%	0.9793
Complex-Network Regression Splines	0.989	0.979	6.119	11.731%	0.9795

* Relative mean square error. ** Relative absolute error. Cases shown in **bold** font indicate best determination models.

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
