# Peer review of "Modeling and Forecasting the COVID-19 Temporal Spread in Greece: An Exploratory Approach Based on Complex Network Defined Splines"

_ijerph, 2020, doi:10.3390/ijerph17134693_

Round 1

Reviewer 1 Report

This is avery interesting paper that examines various mathematical models derived from the existing time trends of COVID 19 infection, along with death rates, admissions to ICU and recoveries, and the relationship between these groups. it is of potential value in optimizing the ability of public health experts and Governments to predict the patterns and dynamics of spread of infection, ICU admissions and deaths enabling timely decisions of the implementations of various preventive measures and resource allocation. The main weakness of the study is that it is based solely on the COVID data from Greece which has been relatively in containing the spread of virus, and where ICU admissions and deaths have been restricted to relatively low proportions of the infected population. The application of the model proposed has not been tested in other countries and may therefore not be applicable. This should be acknowledged.

The English is a bit cumbersome and in some places the grammar is incorrect.

Author Response

Dear Respected Reviewer

Thank you very much for giving us an opportunity to revise our paper. We are grateful to you for your positive and constructive comments and suggestions on our paper. Those comments are all valuable and very helpful for revising and improving our manuscript. We have carefully revised the paper by following your comments.

Cordially

Konstantinos Demertzis, Dimitrios Tsiotas and Lykourgos Magafas

===========================================

This is a very interesting paper that examines various mathematical models derived from the existing time trends of COVID 19 infection, along with death rates, admissions to ICU and recoveries, and the relationship between these groups. It is of potential value in optimizing the ability of public health experts and Governments to predict the patterns and dynamics of the spread of infection, ICU admissions, and deaths enabling timely decisions of the implementations of various preventive measures and resource allocation.

Thank you for the remarks.

The main weakness of the study is that it is based solely on the COVID data from Greece which has been relatively in containing the spread of the virus, and where ICU admissions and deaths have been restricted to relatively low proportions of the infected population. The application of the model proposed has not been tested in other countries and may therefore not be applicable. This should be acknowledged.

We would like to thank the reviewer for this constructive comment that gives us the chance to clarify things further. Our main concern is to prove that this model produces remarkable results based on realistic scenarios. The continuation of our research will focus on tests in data from other countries where have been more hardly affected by the pandemic, such as Italy, Spain, etc. We have discussed this important matter thoroughly in the sections “Conclusions” that we have added two new sub-sections “Limitations” and “Future Research” according to the reviewer’s comment and suggestion.

The English is a bit cumbersome and in some places the grammar is incorrect.

We have rearranged the entire paper, have corrected the typos and grammar errors, and have improved the usage of the English language of the entire manuscript. The paper reads much better now, and the work presented has improved to a level acceptable for the readership and the scientific standing of this journal.

Reviewer 2 Report

The authors are providing an interesting topic related to COVID19 spreading in a timely manner. The paper utilized analysis on time-series that focused on the disease spreading in Greece, taking into account the network effect through non-linear fashions. A unique feature of this paper is that, Greece is a good example of timely response of government policy in reducing the spreading speed, which turned out to be effective in keeping both the infections and deaths at relatively low levels. The paper reveals important policy implications through the curve-fitting modeling, and helps provide insight on the disease control aspect on the COVID19 pandemic.

The followings are several of my comments:

  1. I’m not sure if the authors have used the lagged version of cumulative cases and death in the modeling, but in table 1, where the authors present Pearson’s bivariate correlation analysis, it seems that the one-day lag of both cumulative infected cases and cumulative deaths could serve better the purpose to show the moving trend of increasing cases as well as the correlation. The current X2 and X3 variables may bring in autocorrelations of the daily incidence to the full-period accumulation up to that day by nature of the time series data. If they did, it may be better to label the variables as lagged, for clarification purposes.
  2. The latest retrospective tracing is up to April 18, 2020 in this paper. Since it is almost late of June now, extending the data to early or mid of May could be better. Now that April 21 witnessed a re-surge on the infection, and it would be interesting to see if the model still holds in general.
  3. The simplicity criterion on both the number of terms and the polynomial degree sounds very reasonable, and selecting splines by different knots are understandable. From a statistical point of view, on the other hand, I’m quite curious on how would you interpret the selection of degrees-of-freedom work on the spline modeling. Namely, how did you select the degrees of freedom on the spline terms, especially in the complex-network regression?
  4. The R-squared index has been widely used to help judge model fitting. Since the authors have adopted spline term, have the authors thought about also provide the AUC statistics (which is area under the ROC curve) for model fitting justifications?

Author Response

Dear Respected Reviewer

Thank you very much for giving us an opportunity to revise our paper. We are grateful to you for your positive and constructive comments and suggestions on our paper. Those comments are all valuable and very helpful for revising and improving our manuscript. We have carefully revised the paper by following your comments.

Cordially

Konstantinos Demertzis, Dimitrios Tsiotas and Lykourgos Magafas

==========================================

The authors are providing an interesting topic related to COVID19 spreading in a timely manner. The paper utilized analysis on time-series that focused on the disease spreading in Greece, taking into account the network effect through non-linear fashions. A unique feature of this paper is that, Greece is a good example of timely response of government policy in reducing the spreading speed, which turned out to be effective in keeping both the infections and deaths at relatively low levels. The paper reveals important policy implications through the curve-fitting modeling, and helps provide insight on the disease control aspect on the COVID19 pandemic.

Thank you for the remarks.

The followings are several of my comments:

Q1. I’m not sure if the authors have used the lagged version of cumulative cases and death in the modeling, but in table 1, where the authors present Pearson’s bivariate correlation analysis, it seems that the one-day lag of both cumulative infected cases and cumulative deaths could serve better the purpose to show the moving trend of increasing cases as well as the correlation. The current X2 and X3 variables may bring in autocorrelations of the daily incidence to the full-period accumulation up to that day by nature of the time series data. If they did, it may be better to label the variables as lagged, for clarification purposes.

A1. We would like to thank the reviewer for this constructive comment that contributes to place the Pearson’s bivariate correlation analysis within the framework of time-series analysis. In response to this comment, a relevant commentary was included in the revision as follows: “Towards the first direction, a Pearson’s bivariate correlation analysis is applied to the set of the available variables (X2: X8), and the results are shown in Table 1. In terms of time-series analysis …, computing the Pearson’s bivariate correlation coefficients for variables x, y is equivalent to a cross-correlation analysis with a zero lag (h=0) applied between variables xt and yt+h. Therefore, the first structural perspective does not build on a lagged consideration of the available data since there is neither theoretical evidence in epidemiologic studies … nor any indication based on data observation that the available time-series have periodical structure. As it can be observed in Table 1,…”. Also, a relevant comment was added in the footnoted of Table 1.

Q2. The latest retrospective tracing is up to April 18, 2020 in this paper. Since it is almost late of June now, extending the data to early or mid of May could be better. Now that April 21 witnessed a re-surge on the infection, and it would be interesting to see if the model still holds in general.

A2. We would like to thank the reviewer for this constructive comment that gives us the chance to clarify things further. Our main concern is to prove that this model produces remarkable results based on realistic scenarios. We submitted the paper in the IJERPH Journal at the 2020-05-03 and now have the first round of the revision. We have chosen this Journal because that has a very fast review process but with the Covid-19 pandemic, it seems to have a sort of shortage of researchers available to review manuscripts. This is completely understandable, once reviewers usually are fully busy with other activities. In this situation, as you know it is very difficult we rearrange all the manuscript from scratch in order to update the data and information of the paper (sections will need to change are 1. Introduction, 3. Descriptive analysis of COVID-19 in Greece, 4. Methodology and DATA, 5. Results and Discussion). This is unfair. On the other hand, the continuation of our research will focus on tests in updated data from Greece and other countries where have been more hardly affected by the pandemic, such as Italy, Spain, etc. We have discussed this important matter thoroughly in the sections “Conclusions” that we have added two new sub-sections “Limitations” and “Future Research”.

Q3. The simplicity criterion on both the number of terms and the polynomial degree sounds very reasonable, and selecting splines by different knots are understandable. From a statistical point of view, on the other hand, I’m quite curious on how would you interpret the selection of degrees-of-freedom work on the spline modeling. Namely, how did you select the degrees of freedom on the spline terms, especially in the complex-network regression?

A3. Thank you for this helpful comment. In order to avoid as for global high degree polynomial that the tail wags a lot, two extra constraints have been added at the boundaries (on each end). The constraints make the function extrapolate linearly beyond the boundary knots. With these constraints, the function goes off linearly beyond the range of the data. They then free up a few parameters so the degrees of freedom are always the number of terms that go into the formula minus one (the intercept). The degrees of freedom is therefore the number of predictors that have non-zero coefficients in the model. The spline with K knots has K degrees of freedom because we get back two degrees of freedom for the two constraints on each of the boundaries. The proposed model builds on the recent work of Tsiotas and Magafas (2020), which is published in Physics (MDPI) and introduces a novel approach for the determination of the spline knot vector, based on complex network analysis based on the COVID-19 infection curve of Greece. Generally, the complex-network-based definition of the knot vector offers a missing conceptualization to the splines knots, defining them as borderline points of connectivity of the modularity-based communities. According to this approach, the spline is divided into five knots that represent the evolution of the disease in Greece that went through five stages of declining dynamics. A detailed explanation has added in section 4.2. Regression Splines.

A4. The R-squared index has been widely used to help judge model fitting. Since the authors have adopted spline term, have the authors thought about also provide the AUC statistics (which is area under the ROC curve) for model fitting justifications?

A4. Thank you for this constructive comment. The AUC metrics added in Table 2 in order to assess the model's performance according to the reviewer’s comment and suggestion. Details about the way in which the AUC can be calculated can be read it in the appropriate reference that we added.